# Reciprocal interactions between alveolar progenitor dysfunction and aging promote lung fibrosis

**Jiurong Liang[1†], Guanling Huang[1†], Xue Liu[1], Ningshan Liu[1], Forough Taghavifar[1], Kristy Dai[1], Changfu Yao[1], Nan Deng[2], Yizhou Wang[2], Peter Chen[1], Cory Hogaboam[1], Barry R Stripp[1], William C Parks[1,3], Paul W Noble[1]\*, Dianhua Jiang[1,3]\***

[1]Department of Medicine and Women's Guild Lung Institute, Cedars-Sinai Medical Center, Los Angeles, United States; [2]Genomics Core, Cedars-Sinai Medical Center, los Angeles, United States; [3]Department of Biomedical Sciences, Cedars-Sinai Medical Center, Los Angeles, United States

**\*For correspondence:**
paul.noble@cshs.org (PWN);
dianhua.jiang@cshs.org (DJ)

[†]These authors contributed equally to this work

## Abstract

Aging is a critical risk factor in idiopathic pulmonary fibrosis (IPF). Dysfunction and loss of type 2 alveolar epithelial cells (AEC2s) with failed regeneration is a seminal causal event in the pathogenesis of IPF, although the precise mechanisms for their regenerative failure and demise remain unclear. To systematically examine the genomic program changes of AEC2s in aging and after lung injury, we performed unbiased single-cell RNA-seq analyses of lung epithelial cells from uninjured or bleomycin-injured young and old mice, as well as from lungs of IPF patients and healthy donors. We identified three AEC2 subsets based on their gene signatures. Subset AEC2-1 mainly exist in uninjured lungs, while subsets AEC2-2 and AEC2-3 emerged in injured lungs and increased with aging. Functionally, AEC2 subsets are correlated with progenitor cell renewal. Aging enhanced the expression of the genes related to inflammation, stress responses, senescence, and apoptosis. Interestingly, lung injury increased aging-related gene expression in AEC2s even in young mice. The synergistic effects of aging and injury contributed to impaired AEC2 recovery in aged mouse lungs after injury. In addition, we also identified three subsets of AEC2s from human lungs that formed three similar subsets to mouse AEC2s. IPF AEC2s showed a similar genomic signature to AEC2 subsets from bleomycin-injured old mouse lungs. Taken together, we identified synergistic effects of aging and AEC2 injury in transcriptomic and functional analyses that promoted fibrosis. This study provides new insights into the interactions between aging and lung injury with interesting overlap with diseased IPF AEC2 cells.

## Editor's evaluation

These findings demonstrate the relationship between lung injury, aging, and lung fibrosis with strong evidence that aging modulates the response to lung injury and wound healing. This work furthers our understanding of fibrosis in humans, where idiopathic pulmonary fibrosis occurs most commonly at an advanced age.

## Introduction

Type 2 alveolar epithelial cells (AEC2s) function as progenitor cells that maintain epithelial homeostasis and repair damaged epithelium after lung injury (*Barkauskas et al., 2013*; *Desai et al., 2014*; *Hogan et al., 2014*; *Jiang et al., 2020a*). Distal lung epithelial progenitor cell function declines (*Watson*

*et al., 2020*) and gene expression profiles change in AEC2s with aging (*McQuattie-Pimentel et al., 2021*).

Aging is an important risk factor in idiopathic pulmonary fibrosis (IPF) (*Hecker, 2018*; *Hecker et al., 2014*; *Thannickal, 2013*). The incidence, prevalence, and mortality of IPF all increase with age (*Raghu et al., 2016*; *Rojas et al., 2015*). Growing evidence suggests that IPF is a result of alveolar epithelial dysfunction and inadequate regenerative capacity that leads to basal-like cell expansion and excessive fibroblast activation with matrix deposition and destruction of the normal lung architecture (*Borok et al., 2020*; *Cho and Stout-Delgado, 2020*; *Noble et al., 2012*; *Schneider et al., 2021*). Both the numbers and progenitor cell renewal capacity of AEC2s in IPF lungs are signfinicantly reduced (*Liang et al., 2016*; *Xu et al., 2016*). Phenotypes of cellular aging in AEC2s including mitochondrial dysfunction (*Bueno et al., 2018*), senescence (*Jiang et al., 2017*; *Yao et al., 2021*), endoplasmic reticulum (ER) stress (*Burman et al., 2018*), and premature lung aging (*Chilosi et al., 2013*) have been described in IPF. However, the interplay between aging, injury, and AEC2 progenitor dysfunction and renewal in IPF are not fully understood.

Heterogeneity in distal lung epithelial cells has been increasingly recognized in recent studies. Subclusters of alveolar epithelial cells are distributed differently between human fibrotic and healthy donor lungs (*Adams et al., 2020*; *Habermann et al., 2020*; *Reyfman et al., 2019*; *Yao et al., 2021*). AEC2 subsets were reported in mouse lungs after lipopolysaccharides (LPS) injury (*Jiang et al., 2020b*; *Riemondy et al., 2019*), following pneumonectomy (*Wu et al., 2020*), as well as with TiO₂ exposure (*Joshi et al., 2020*). Subclusters of Krt8-positive AEC2s (*Strunz et al., 2020*) and CLDN4-expressing AEC2s were reported in IPF (*Strunz et al., 2020*) and bleomycin-injured mouse lungs (*Choi et al., 2020*). KRT5⁻/KRT17⁺ cells accumulated in human lungs with pulmonary fibrosis (*Habermann et al., 2020*). However, there is a paucity of comprehensive studies examining AEC2 subsets and their effector functions in the context of the combination of aging, lung injury, and lung fibrosis.

In the current study, we took an unbiased approach – single-cell RNA-sequencing (scRNA-seq) of primary flow cytometry-enriched lung epithelial cells – to systematically investigate the genetic signatures and programs of AEC2s in young and old mouse lungs under homeostasis and after experimental fibrotic lung injury, and comparatively examined these genetic programs in AEC2s from IPF and heathy donor lungs. Interestingly, we identified three subsets of AEC2s based on gene expression profiling. The genomic programming changes of AEC2 subsets were correlated with lung injury and influenced by aging. Most importantly, our data revealed a previously unrecognized interaction between aging and AEC2 injury which contributed to impaired AEC2 progenitor function. Furthermore, AEC2s from IPF lungs have similar gene signatures as AEC2s from bleomycin-injured old but not young mouse lungs.

## Results
### Defining epithelial cell transcriptome profiles in young and old mouse lungs

To better focus our investigation on the genetic signatures and programs of lung alveolar epithelial cells during aging and following lung injury, we performed scRNA-seq on flow sorted epithelial cells (EpCAM⁺CD31⁻CD34⁻CD45⁻) from lungs of uninjured (day 0), 4, 14, and 28 days post bleomycin injury of young and old mice (*Figure 1A*). A total of 96,213 cells were analyzed, and the cells from 26 individual mice overlapped very well (*Figure 1B*). The major lung epithelial cell types, AEC2, AEC1, basal cells, club cells, ciliated cells, pulmonary neuroendocrine cells, and proliferative cells were readily identified with canonical cell markers (*Figure 1C–E*). The AEC2 cluster was the largest among the lung epithelial cell populations (*Figure 1C*). The general aging marker genes, including *B2M*, *H2-K1*, *H2-D1*, *H2-Q7*, and *CHIA1* were all upregulated in AEC2s from uninjured old mice (*Figure 1F, G*), consistent with previous reports (*Angelidis et al., 2019*; *López-Otín et al., 2013*).

### AEC2 subsets with aging and lung injury

AEC2 heterogenity following lung injury in mouse models has been extensively reported (*Joshi et al., 2020*; *Kobayashi et al., 2020*; *Riemondy et al., 2019*; *Strunz et al., 2020*; *Wu et al., 2020*). However, there is an incomplete understanding of AEC2 subsets under the influence of both aging and lung injury. We therefore analyzed the gene expression of a total of 57,717 AEC2s from both

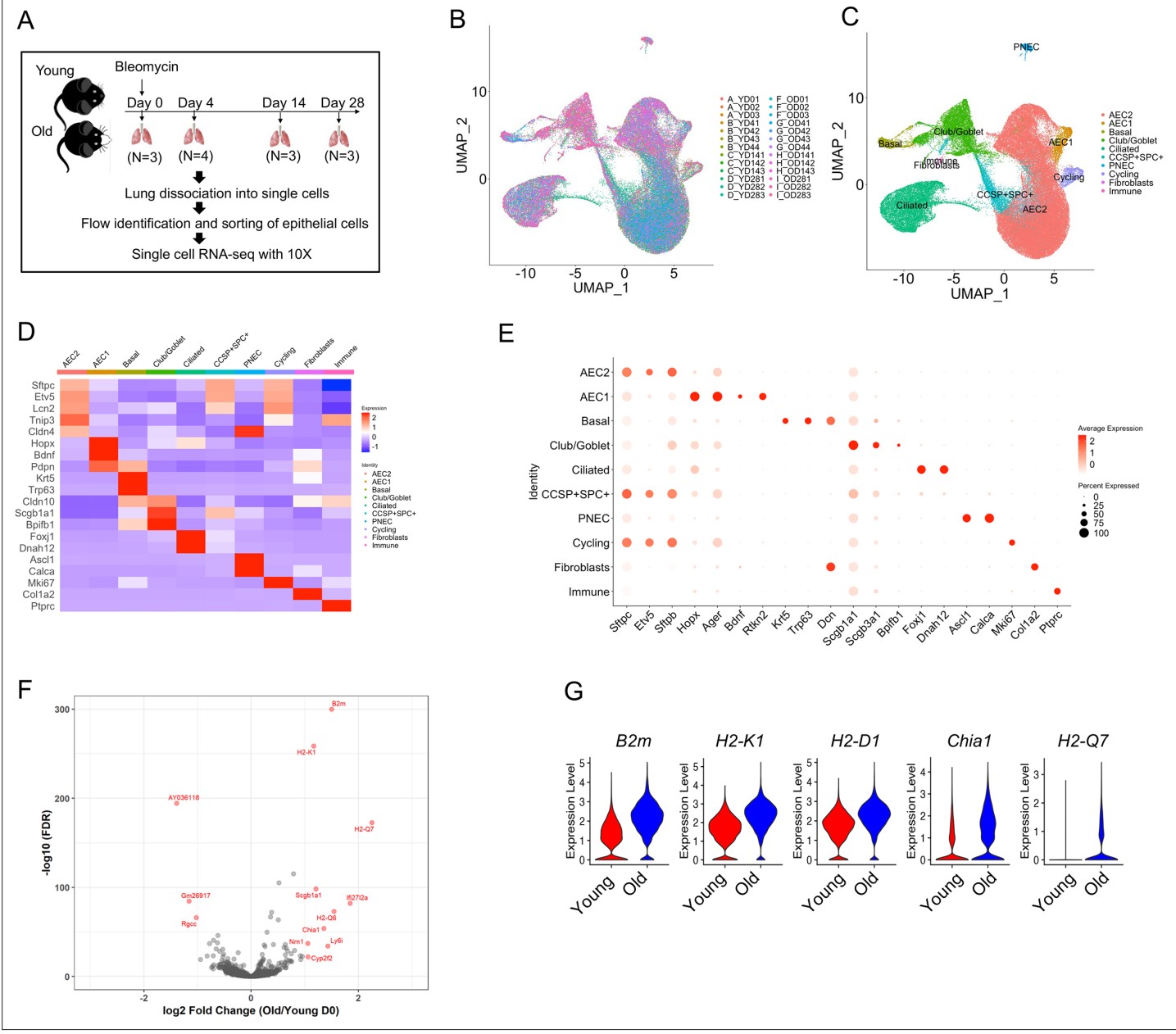

**Figure 1.** Transcriptome profiles of lung epithelial cells in young and old mice. (**A**) Schematic of scRNA-seq analysis of flow sorted EpCAM+CD31−CD34−CD45− cells from lungs of uninjured (day 0) (*n* = 3), day 4 (*n* = 4), day 14 (*n* = 3), and day 28 post injury (*n* = 3) young and old mice. (**B**) Uniform Manifold Approximation and Projection (UMAP) visualization of 96,213 cells from all 26 samples. (**C**) UMAP visualization of epithelial cell clusters. (**D**) Heatmap of epithelial cell clusters. (**E**) Dot plots of conventional marker genes of epithelial cell clusters. (**F, G**) Gene expression in type 2 alveolar epithelial cells (AEC2s) from old vs young uninjured mice.

homeostatic and bleomycin-injured young and old mouse lungs, and identified three AEC2 subsets, AEC2-1, AEC2-2, and AEC2-3 according to their gene expression signatures (***Figure 2A, B***). AEC2-2 showed the lowest correlation silhouette value suggesting an intermediate status between AEC2-1 and AEC2-3 (***Figure 2C***). Pseudotime analysis identified the AEC2-1 subset with the lowest entropy and AEC2-3 with the highest entropy, suggesting a transition from AEC2-1 to AEC2-2 and further to AEC2-3 (***Figure 2D***). Correlation spanning tree analysis supported these sequential dynamics (***Figure 2D***). Furthermore, RNA trajectory analysis using a cell lineage and pseudotime inference tool, Slingshot (***Street et al., 2018***), confirmed that subset AEC2-3 cells were from subset AEC2-1 via

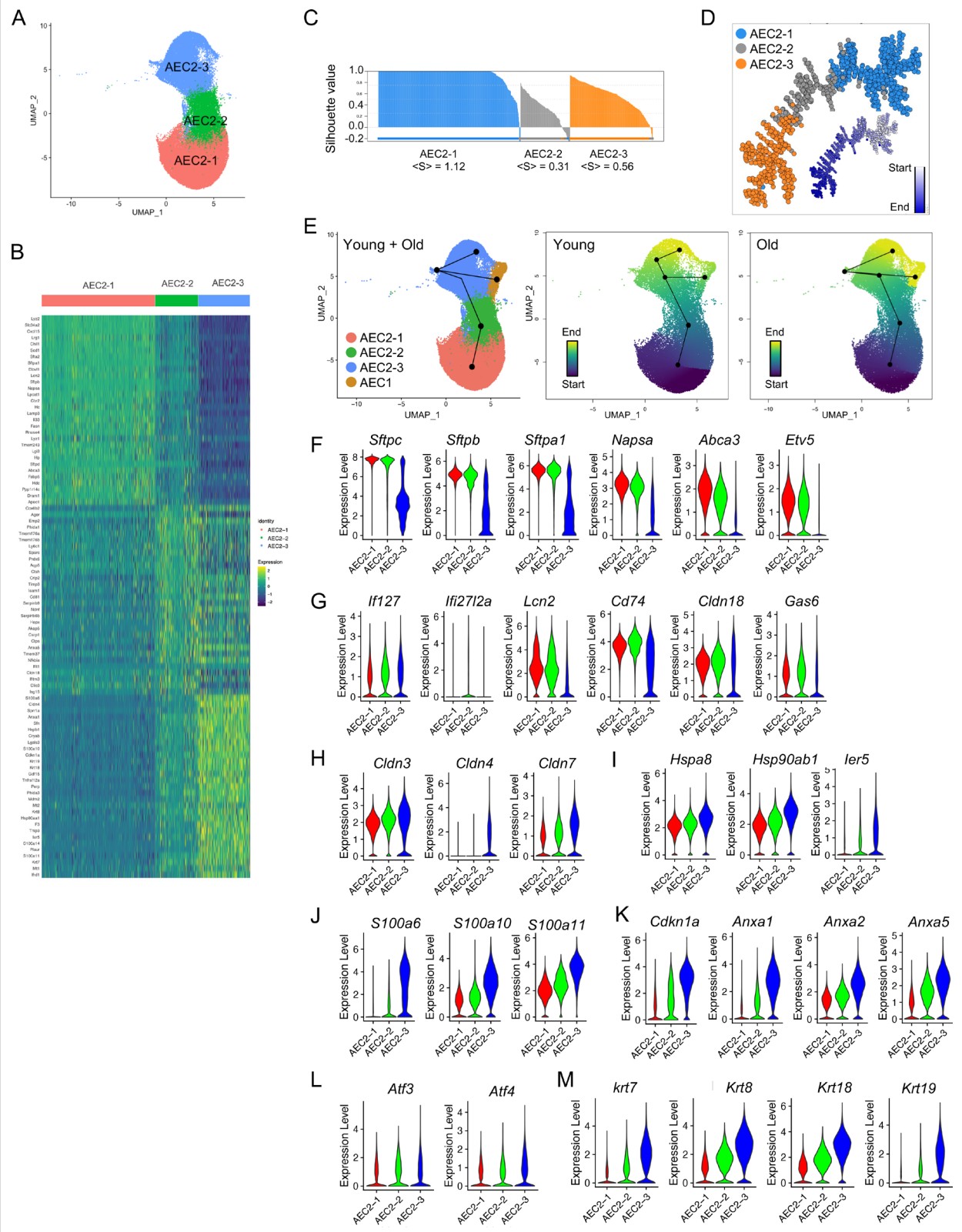

**Figure 2.** Definition of type 2 alveolar epithelial cell (AEC2) subsets. (**A**) UMAP of 57,717 AEC2s showing in three subsets of AEC2s (red, AEC2-1; green, AEC2-2; and blue, AEC2-3). (**B**) Heatmap representing characteristics of three subsets of AEC2s. Each column represents the average expression value for one cell, grouped by cell cluster. Gene expression values are normalized in rows. (**C**) Correlation silhouette of AEC2 subsets. (**D**) Pseudotime analysis and correlation spanning tree of AEC2 subsets. (**E**) Slingshot trajectory inference analysis showed the lineage reconstructions of the AEC2 subsets

*Figure 2 continued on next page*

Figure 2 continued

and AEC1 clusters from young and old lungs. (F–M) Violin plots of gene expression in AEC2 subsets (red, AEC2-1; green, AEC2-2; and blue, AEC2-3). Expression of subset AEC2-1 marker genes (F), expression of subset AEC2-2 marker genes (G), expression of claudin family genes (H), expression of heat shock protein family genes (I), expression of S100 protein family genes (J), expression of *Cdkn1*a and annexin family genes (K), expression of *Atf3* and *Atf4* (L), and expression of the keratin family genes (M).

subset AEC2-2, and AEC1 cells were from subset AEC2-3 cells in both young and old mice lungs, although further heterogeneity existed in AEC2-3 clusters (*Figure 2E*).

Subset AEC2-1 cells express typical signature genes of AEC2s including *Sftpc*, *Sftpb*, *Sftpa1*, *Napsa*, *Abca3*, and *Etv5*. All these AEC2 marker genes were significantly downregulated in subsets AEC2-3 (*Figure 2F*).

Increased interferon (IFN) signaling was observed in AEC2-2 cells with increased expression of IFN-induced genes, *Ifi27* and *Ifi2712a* (*Figure 2G*). Other genes upregulated in subset AEC2-2 cells included *Lcn2*, *Cd74*, *Cldn18*, *Bcam*, and *Gas6* (*Figure 2G*).

Several gene families were upregulated in subset AEC2-3. *Cldn4* was expressed only in AEC2-3 cells, while *Cldn3* and *Cldn7* were expressed in all three subsets of AEC2s with highest expression in AEC2-3 (*Figure 2H*). Gene families include the heat shock protein family (*Figure 2I*), S100 proteins (*Figure 2J*), senescence and apoptosis family (*Figure 2K*), endoplasmic reticulum (ER) stress-related genes (*Figure 2L*), and keratin genes (*Figure 2M*) were all upregulated in subset AEC2-2 and their expression was further elevated in subset AEC2-3.

Pathway analyses idenfitified that both subsets AEC2-2 and AEC2-3 showed upregulated oxidative phosphorylation and NRF2-mediated oxidative stress response genes, compared to subset AEC2-1 (*Supplementary file 1a, b*). The EIF2 signaling, eIF4 and p70S6K signaling, and mTOR signaling pathways were upregulated in subsets AEC2-2 and AEC2-3 (*Supplementary file 1a, b*). In addition, both subset 2 and 3 AEC2s showed increased unfolded protein response, p53 signaling, and apoptosis signaling pathways relative to subset 1 (*Supplementary file 1a, b*). IFN signaling was upregulated in subset AEC2-2 (*Supplementary file 1a*). These data suggest that the cells in AEC2-2 and AEC2-3 were demonstrating stress responses.

The gene signature and signaling pathways of these three AEC2 subsets indicated that AEC2s in subset AEC2-1 were intact, homeostatic AEC2s, while subset AEC2-2 and subset AEC2-3 were injured AEC2s.

## Dynamic of AEC2 subsets in aging and lung injury

Next, we investigated how aging and lung injury affected the subsets of AEC2s. Four days post bleomycin injury is the time point with maximum AEC2 injury, and by days 14 and 28 post injury there is substantial AEC2 recovery (*Liang et al., 2016*). Over 80% of total AEC2s in uninjured young mouse lungs were subset AEC2-1 cells and the percentage of AEC2-1 cells was slightly lower in the uninjured old mouse lungs (*Figure 3A, B*). At day 4 post bleomycin injury, AEC2s shifted from subset AEC2-1 to AEC2-2 and AEC2-3 in both young and old mouse lungs. It is interesting that at day 14 post bleomycin injury, during recovery, the intact AEC2s in AEC2-1 subset were partially recovered in young mouse lungs (from 29.05% at day 4 to 60.46% at day 14 post bleomycin injury), but the old mouse lungs were continuously losing cells in the AEC2-1 subset (from 41.44% at day 4 to 18.83% at day 14 post bleomycin injury). The majority of AEC2s in old mouse lungs at day 14 post bleomycin injury were subset AEC2-3 cells (*Figure 3A, B*). There was some degree of recovery of subset AEC2-1 in old mouse lungs at day 28 compared to day 14 post bleomycin injury (*Figure 3A, B*). However, the old mouse lungs continued to have much lower percentages of AEC2-1 (old 34% vs young 54%) and higher percentages of AEC2-3 (old 46% vs young 12%) at day 28 relative to that of the lungs from young mice (*Figure 3B*). These data suggest that aging affects AEC2 subset evolution in the lung with a reduced subset of intact AEC2s and increased proportions of damaged AEC2s in old mouse lungs, especially after injury.

We then interrogated the progenitor functions of the AEC2 subsets we identified. We flow sorted primary AEC2s (*Liang et al., 2022*; *Liang et al., 2016*) from uninjured and day 4 bleomycin-injured young mice and applied the cells to 3D orgnoid cultures. The renewal capacity of AEC2s was measured by colony-forming efficiency (CFE). AEC2s from bleomycin-injured lungs had decreased regenerative capacity compared to AEC2s from uninjured lungs as demonstrated by decreased CFEs (*Figure 3C*).

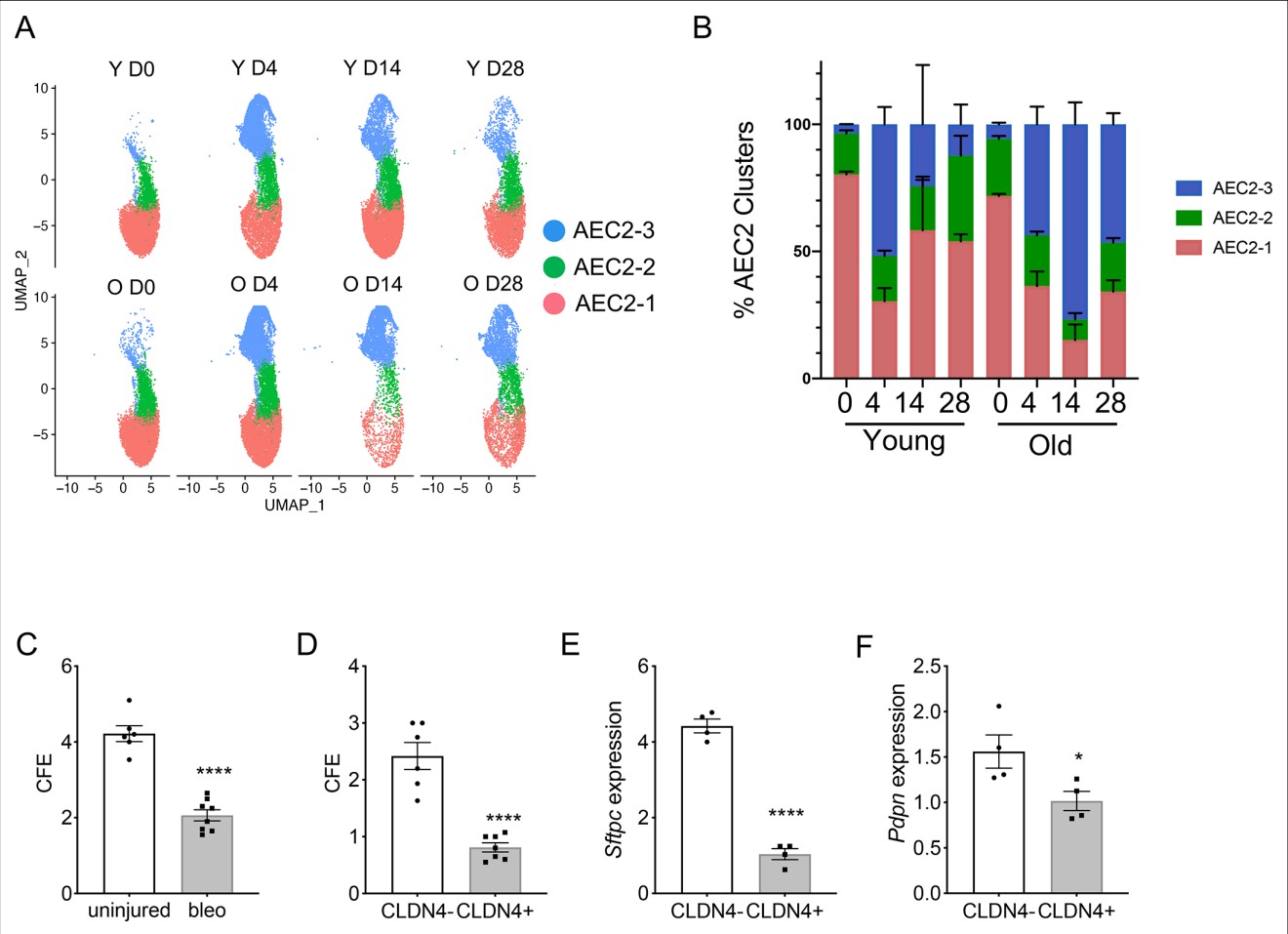

**Figure 3.** Type 2 alveolar epithelial cells (AEC2s) subsets were influenced by both aging and lung injury. (**A**) UMAP showing the distributions of the three subsets of AEC2s grouped by age and injury date (red, AEC2-1; green, AEC2-2; and blue, AEC2-3. Y = young; O = old). (**B**) Percentage of AEC2 subsets grouped by age and injury date (red, AEC2-1; green, AEC2-2; and blue, AEC2-3). (**C**) Colony-forming efficiency (CFE) of AEC2s from uninjured and day 4 bleomycin-injured young mice ($n = 6–8$, ****$p < 0.0001$). (**D**) CFE of CLDN4+ and CLDN4− AEC2s isolated at day 14 after bleomycin treatment ($n = 7–8$, ****$p < 0.0001$). *Sftpc* (**E**) and *Pdpn* (**F**) expression of CLDN4+ and CLDN4− AEC2s derived from 3D cultured organoids by reverse transcription-polymerase chain reaction (RT-PCR) ($n = 4$, ****$p < 0.0001$, *$p < 0.05$). (C-F) Statistical anayses were by unpaired two-tailed Student's *t*-test.

Since Claudin-4 (CLDN4) was only expressed in subset AEC2-3, we used CLDN4 as a cell surface marker to flow sort subset AEC2-3 cells (EpCAM⁺CD31⁻CD34⁻CD45⁻CD24⁻Sca-1⁻CLDN4⁺) from mouse lungs at day 14 after bleomycin injury and compared the CFEs to AEC2s in subset AEC2-1 and AEC2-2 (EpCAM⁺CD31⁻CD34⁻CD45⁻CD24⁻Sca-1⁻CLDN4⁻). Our results showed that CLDN4-positive AEC2s had significantly reduced regenerative capacity relative to that of CLDN4-negative AEC2s (*Figure 3D*). We further showed that AEC2s derived from CLDN4-positive organoids had decreased *Sftpc* and *Pdpn* expression compared to that of AEC2s form CLDN4-negative organoids by RT-PCR (*Figure 3E, F*), suggesting decreased AEC2 integrity and differentiation capacity of CLND4-positive AEC2-3 cells.

## Aging enhanced injury-related gene expression in AEC2s

The altered AEC2 subsets in bleomycin-injured aged mouse lungs suggested aging has an important impact on gene expression and function of AEC2s during lung injury. To gain further insights into the impact of aging on AEC2 injury and repair, we compared gene expression in AEC2s from young and old mice at multiple time points after bleomycin treatment. As expected, bleomycin injury increased the expression of inflammation and oxidative stress-related genes (*Figure 4A*), S100 family genes (*Figure 4B*), heat shock protein family genes (*Figure 4C*), senescence (*Figure 4D*) and apoptosis (*Figure 4D*), and ER stress-related genes (*Figure 4F*). Interestingly, aged AEC2s showed lower

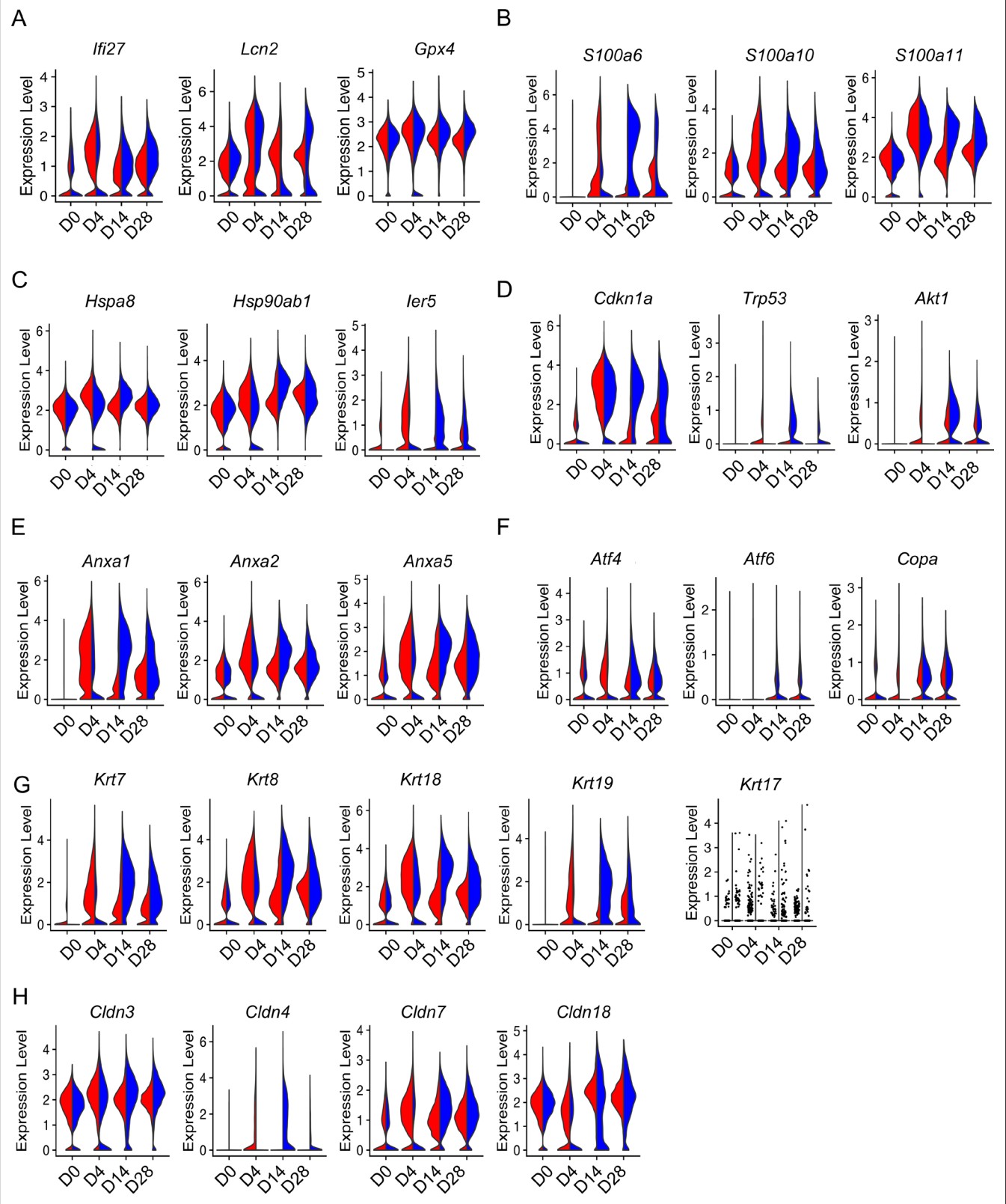

**Figure 4.** Aging enhanced type 2 alveolar epithelial cell (AEC2) injury with bleomycin time course. Violin plots of gene expression in AEC2 from young and old mice at baseline day (D) 0 and days 4, 14, and 28 after bleomycin treatment. (**A**) Inflammatory and oxidative stress-related genes in AEC2s grouped by age and injury date; (**B**) S100a family genes; (**C**) heat shock protein family genes; (**D**) Senescence gene; (**E**) apoptosis-related genes; (**F**) ER stress-related genes; (**G**) keratin family genes; and (**H**) claudin family genes. Red, young; blue, old.

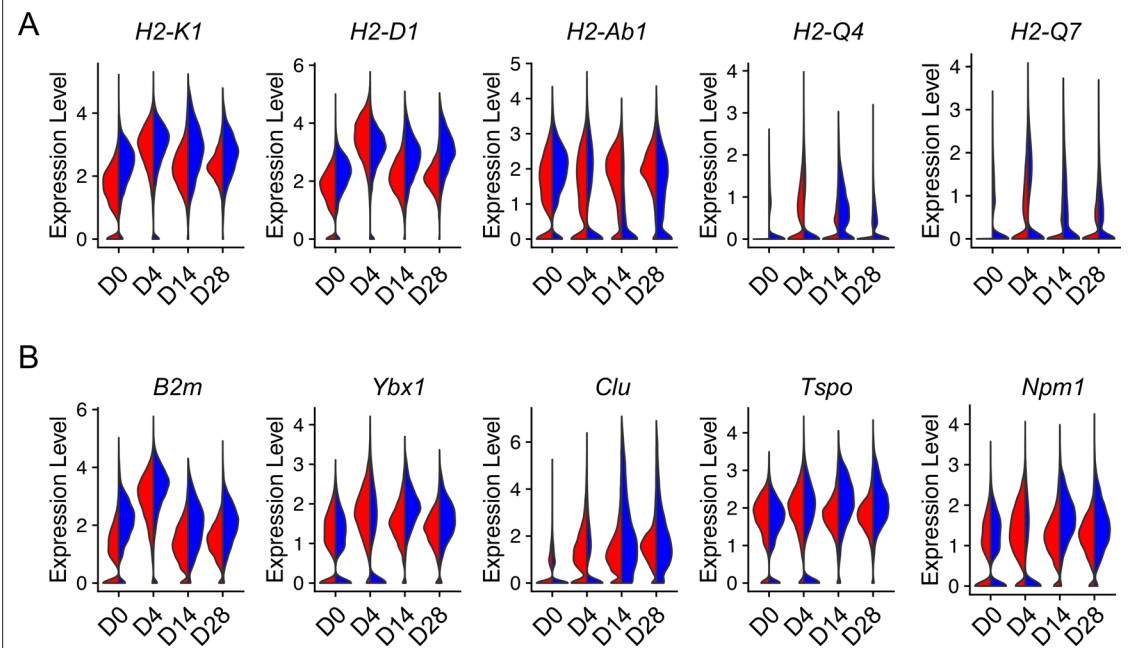

**Figure 5.** Injury-induced aging-related gene expression in type 2 alveolar epithelial cells (AEC2s). Violin plots of aging-related gene expression in AEC2 from young and old mice at baseline day (D) 0 and days 4, 14, and 28 after bleomycin treatment. (**A**) Histocompatibility 2 genes. (**B**) Aging hallmark genes. Red, young; blue, old.

response compared to that of young AEC2s at day 4 after bleomycin treatment, while the expression levels of all these genes were elevated in aged AEC2s compared to that of young AEC2s at the later time points (*Figure 4A–F*), suggesting exaggerated AEC2 injury and delayed recovery with aging.

Reports showed that several AEC2 subsets including Krt8 expressing AEC2s (*Strunz et al., 2020*) and Cldn4-expressing AEC2s (*Kobayashi et al., 2020*) were transitional AEC2s which accumulated in bleomycin-injured mouse lungs. We therefore analyzed the effect of aging on the expression of these genes in AEC2s following bleomycin injury over time. Indeed, the expression of multiple genes in the keratin family including *Krt7*, *Krt8*, *Krt18*, and *Krt19* were all upregulated and further increased in aged AEC2s after bleomycin injury (*Figure 4G*). The expression level of Krt17 was very low in mouse AEC2s but with a trend toward increase with injury and aging (*Figure 4G*). It was reported that Krt17-expressing AEC2s accumulated in IPF lungs (*Habermann et al., 2020*). The expression of keratin family genes peaked at day 4 and declined toward baseline levels at days 14 and 28 in AEC2s from young mice. However, the expression of keratin genes remained high at days 14 and 28 in AEC2s from old mice (*Figure 4G*). Multiple claudin family genes including *Cldn3*, *Cldn7*, and *Cldn8* were upregulated after bleomycin injury and further elevated with aging (*Figure 4H*). *Cldn4* expression was very low in uninjured AEC2s and its expression increased with bleomycin injury and aging (*Figure 4H*). This result is consistent with what we showed in *Figure 2H* that Cldn4 was mainly expressed in subset AEC2-3 cells. The persistently elevated expression of both Krt8 and Cldn4 in AEC2s from aged mice at later time points of bleomycin injury suggested further impaired AEC2 differentiation to AT1 cells with aging after lung injury.

## Injury elevated aging-related gene expression in AEC2s

Lung injury promotes AEC2 senescence which is also a signature of aging (*Lee et al., 2012*). To gain insight into how injury can influence aging-related genetic changes in AEC2s, we analyzed aging hallmark genes expressed in AEC2s from young and old mice after bleomycin injury. Interestingly, the expression of multiple aging-related genes including class II histone protein genes (*Figure 5A*) and other aging hallmark genes including *B2m*, *Ybx1*, *Clu*, *Tspo*, and *Npm1* (*Figure 5B*) were elevated in bleomycin-injured AEC2s from both young and old mice. The expression levels of aging hall-mark genes were equally high in injured young and aged AEC2s at day 4 after bleomycin treatment

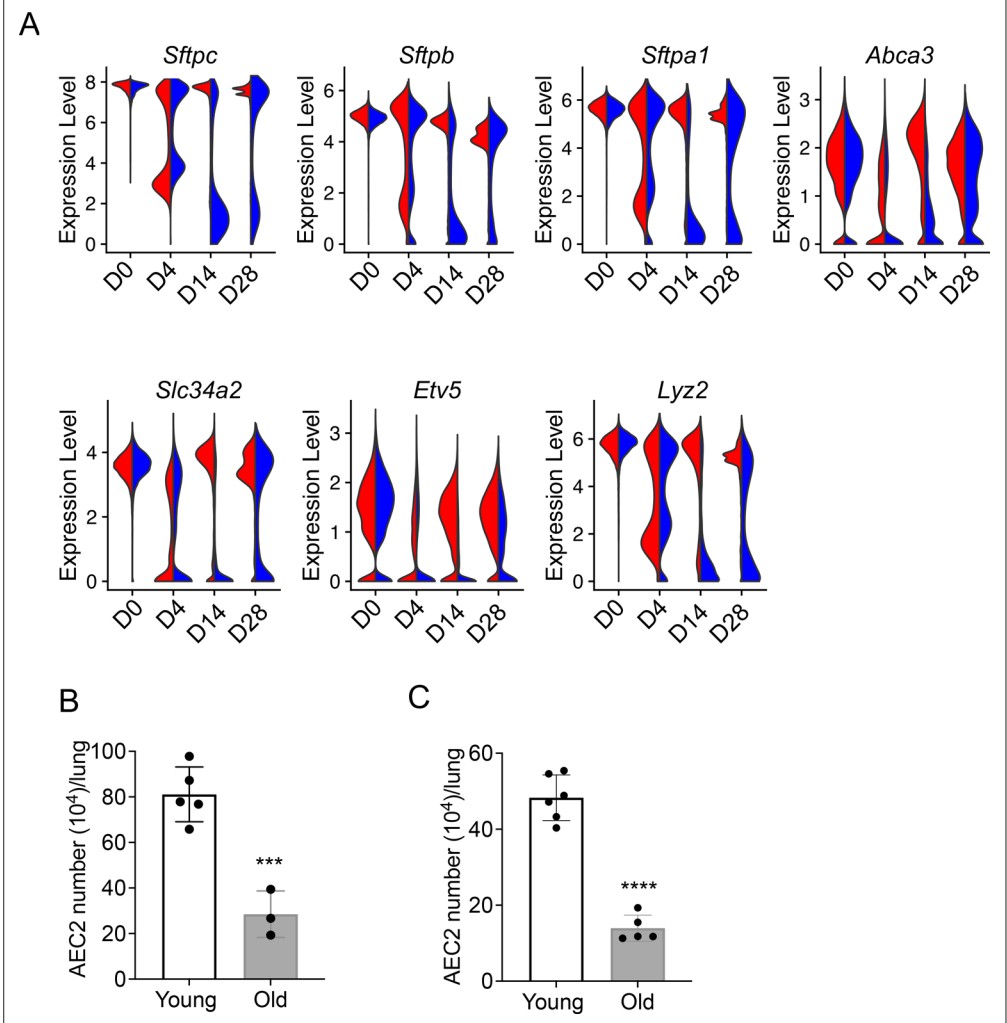

**Figure 6.** Decreased renewal capacity of injured aged type 2 alveolar epithelial cells (AEC2s). (**A**) Violin plots of gene expression in AEC2 from young and old mice at baseline day (D) 0 and days 4, 14, and 28 after bleomycin treatment. Red, young; blue, old. (**B**) Number of AEC2s recovered from uninjured young and old mice ($n$ = 5–6, ***p < 0.001, by unpaired two-tailed Student's $t$-test). (**C**) Number of AEC2s recovered from bleomycin day 28 young and old mouse lungs ($n$ = 3–5, ****p < 0.0001, by unpaired two-tailed Student's $t$-test).

(*Figure 5A, B*). These results suggested that not only aging affects AEC2 function after injury, but injury also skews AEC2s toward aging-like genetic changes.

## Impaired AEC2 recovery in bleomycin-injured aged mouse lungs

Our data suggest that aging enhances inflammatory and stress response, senescence, ER stress, and apoptosis-related gene expression in AEC2s after lung injury, meanwhile injury skewed the gene expression in AEC2s toward aging-like changes. We anticipated that this interaction between aging and injury of AEC2s would impair AEC2 recovery in aged mouse lungs after bleomycin injury. We first examined expression of AEC2 marker genes in AEC2s from young and old mice following bleomycin treatement over time. Multiple surfactant genes including *Sftpc*, *Sftpb*, *Sftpa1*, and several other AEC2 marker genes including *Abca3*, *Slc34a2*, *Lyz2*, and *Etv5* were all downregulated in both young and old AEC2s at day 4, the time point with maximum AEC2 injury after bleomycin treatment. The expression of AEC2 marker genes in AEC2s from young mice were recovered close to baseline levels at days 14 and 28 but their expression in AEC2s from old mice remained reduced even at day 28 (*Figure 6A*). Old mouse lungs contained fewer AEC2s at baseline without injury (*Figure 6B*). The numbers of AEC2s in old mouse lungs at day 28 after bleomycin-injured were further reduced relative

to that of young mice due to impaire AEC2 recovery with aging (**Figure 6B**). The decreased AEC2 recovery in bleomycin-injured old mice was aligned with senescence and impaired progenitor renewal of AEC2s with aging (**Childs et al., 2015**; **Liang et al., 2022**).

## IPF AEC2s showed similar gene signature patterns as injured aged murine AEC2s

It was reported that an AEC2 subset representing transitional AEC2s exist in both bleomycin-injured mouse lungs and human IPF lungs (**Huang and Petretto, 2021**), suggesting a similarity in AEC2 subsets between mouse fibrosis and IPF. Next, we determined if the three AEC2 subsets we identified in the mouse model was relevant to human disease. We analyzed scRNA-seq data of flow cytometry-enriched lung epithelial (EpCAM⁺CD31⁻CD45⁻) cells from lung tissues of six IPF patients (11,381 cells) and six healthy donors (14,687 cells) (**Liang et al., 2022**). IPF AEC2s had a significant decrease in expression of classical AEC2 genes including *SFTPC*, *SLC34A2*, *ABCA3*, and *ETV5*, when compared with AEC2s from healthy donors (**Figure 7A**). IPF AEC2s showed strong IFN signaling with a higher IFN activation score and elevated gene expression of *IFI27* (**Figure 7B**). Genes including *LCN2*, *CD74*, *BCAM*, and *GAS6* were significantly upregulated in IPF AEC2s (**Figure 7C**) which are the same group of genes that were upregulated in mouse subset AEC2-2.

Several ER stress-related genes (**Figure 7D**), claudin family genes (**Figure 7E**), S100 family genes (**Figure 7F**), as well as keratin family genes (**Figure 7G**) were upregulated in IPF AEC2s which are the orthologous genes upregulated in the mouse subset AEC2-3. Among the keratin family genes that were upregulated in IPF lungs, KRT17 was most elevated compared to the low expression level in healthy AEC2s. This finding was consistent with the report that KRT17 expressing cells accumulated in IPF lungs (**Adams et al., 2020**; **Habermann et al., 2020**).

Next we asked if AEC2s from human lungs could be separated into different subsets, similar to the mouse AEC2s. Indeed, three clusters were readily recognized and overlapped well with AEC2s from healthy and IPF lungs (**Figure 7H**). While subset 1 and 2 clusters were predominate in healthy lungs, the subset 3 cluster mainly existed in IPF lungs (**Figure 7I**). Both subsets 1 and 2 expressed the AEC2 marker gene, *SFTPC* (**Figure 7J**). Subset 2 and 3 cluster cells expressed IFN-induced gene, *IFI27* (**Figure 7K**), *GAS6* (**Figure 7L**), and *CLDN4* (**Figure 7M**), which are the orthologous genes that were highly expressed in mouse AEC2 subsets AEC2-2 and AEC2-3 in injured mouse lungs (**Figure 2G, H**). Thus, the AEC2 subset classification also applies to human AEC2s.

To confirm our findings, we performed immunofluorescence staining of human lung sections for CLDN4 and HTII-280. As expected, there was a significant loss of HTII-280⁺ AEC2s in IPF lungs (**Figure 7N**) consistent with our previous reports (**Liang et al., 2016**; **Yao et al., 2021**). Interestingly, the majority of remaining AEC2s in IPF lungs were CLDN4⁺ while only a small portion of AEC2s in healthy lungs were CLDN4⁺ (**Figure 7N**). We quantified the number of total HTII-280⁺ AEC2s and HTII-280⁺CLDN4⁺ AEC2s in multiple sections. The percentage of CLDN4⁺HTII-280⁺ in total HTII-280⁺ AEC2s was much higher in IPF lung sections than that of healthy lung sections (**Figure 7O**).

## Discussion

Lung aging showed the hallmarks of cellular aging including genomic instability, nutrient sensing aberrations, cell–cell communication impairment, mitochondrial abnormalities, aberrant proteostasis, stem cell exhaustion, and cellular senescence (**Cho and Stout-Delgado, 2020**; **López-Otín et al., 2013**; **Oliviero et al., 2022**; **Rojas et al., 2015**; **Thannickal, 2013**). Alveolar epithelial progenitor cell dysfunction is an important pathogenic characteristic in lung aging and aging-associated lung fibrosis, IPF (**Chanda et al., 2021**; **Cho and Stout-Delgado, 2020**; **Cui et al., 2022**; **Liang et al., 2022**; **Selman and Pardo, 2020**). In this study, we analyzed gene signatures and functional alterations of AEC2s with aging during lung injury to illustrate the impact of aging on lung epithelilum injury and repair. Our data demonstrate that aging augmented injury-related gene expression, while lung injury also elevated aging-related gene expression in AEC2s, resulting in a vicious cycle of genetic reprogramming that further impairs AEC2 functions.

The heterogeneity of lung epithelial cells has been examined in both mouse models and human lung diseases with scRNA-seq (**Adams et al., 2020**; **Carraro et al., 2020**; **Habermann et al., 2020**; **Reyfman et al., 2019**; **Strunz et al., 2020**). Most scRNA-seq analyses were performed with the

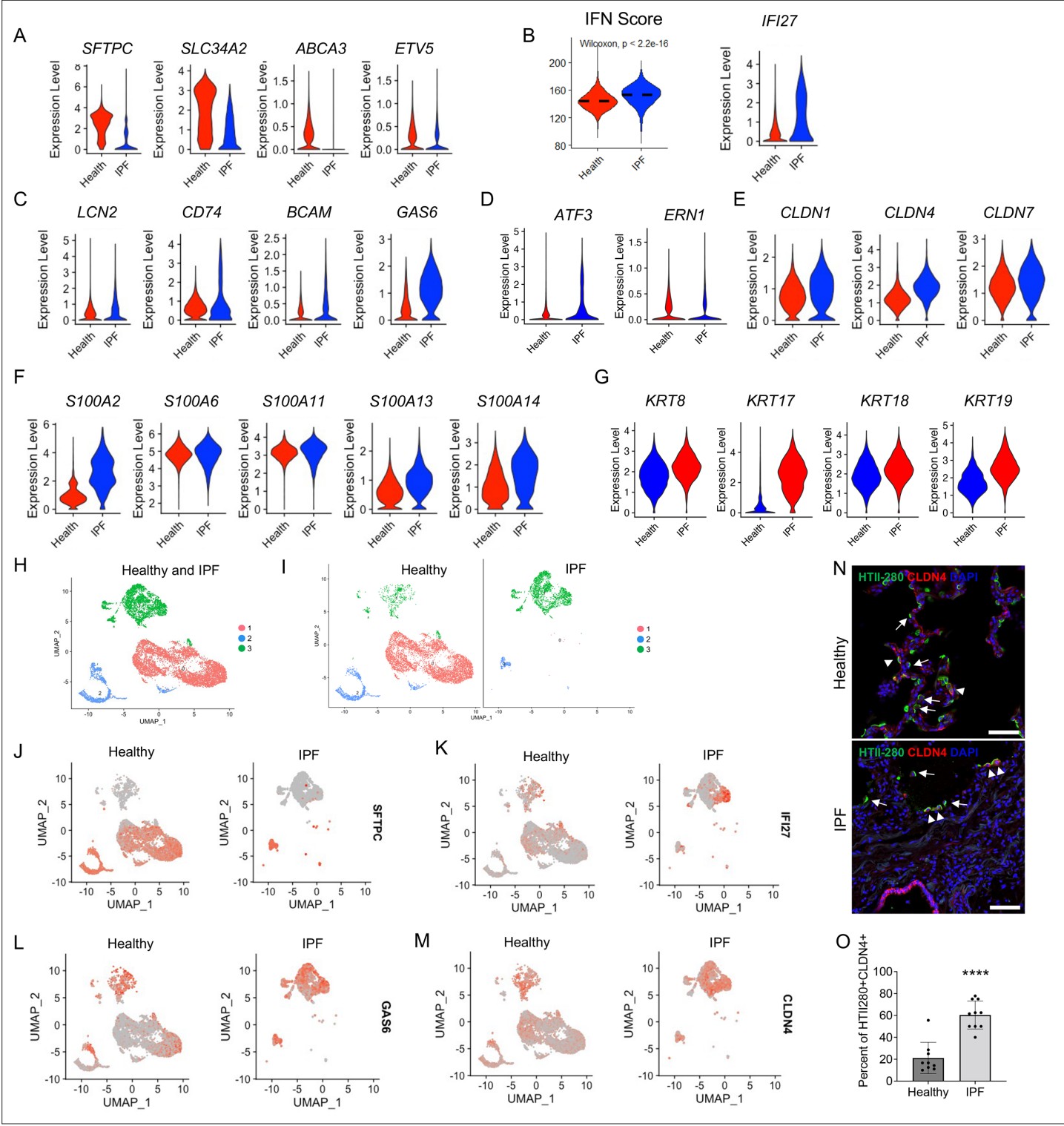

**Figure 7.** Idiopathic pulmonary fibrosis (IPF) type 2 alveolar epithelial cells (AEC2s) share the gene signatures of mouse AEC2 subsets AEC2-2 and AEC2-3. (**A**) Violin plots of expression of mouse AEC2-1 signature genes in healthy and IPF AEC2s (red, healthy; blue, IPF AEC2). (**B**) IFN activation score and expression of *IFI27* of AEC2s from healthy and IPF lungs (red, healthy; blue, IPF). (**C**) Violin plots of expression of mouse AEC2-2 signature genes in healthy and IPF AEC2s (red, healthy; blue, IPF AEC2). Violin plots of expression of AEC2-3 signature genes, including ER stress-related (**D**), claudin family (**E**), S100 family (**F**), and keratin family (**G**) in healthy and IPF AEC2s (red, healthy; blue, IPF). (**H**) UMAP showing three subsets of human AEC2s from healthy and IPF lungs. (**I**) AEC2 subset distribution in healthy and IPF AEC2s. Selected gene expression of AEC2 subsets from healthy and IPF lungs, *SFTPC* (**J**); *IFI27* (**K**); *GAS6* (**L**); and *CLDN4* (**M**). (**N**) HTII-280 (arrows) and CLDN4 co-staining (arrowheads) of human lung sections, scale bars 50 μm. (**O**)

*Figure 7 continued on next page*

*Figure 7 continued*

Quantitation of the percentage of HTII280+CLDN4+ cells in total HTII-280+ AEC2s in healthy and IPF lungs (*n* = 9–10, ****p < 0.0001, by unpaired two-tailed Student's *t*-test).

inclusion of all cell types in the lung. In this study, we performed scRNA-seq with flow-enriched epithelial cells from bleomycin-injured young and aged mouse lungs, as well as lungs from IPF patients and healthy donors. This allowed us to deeply analyze the gene signature of AEC2s under the influence of both aging and injury. We identified three subsets of AEC2s. Subset AEC2-1 cells represent intact, homeostatic AEC2s and subsets AEC2-2 and AEC2-3 represent injured AEC2s.

The distribution of AEC2 subsets in the lung is influenced by aging. At baseline, over 80% of AEC2s are subset AEC2-1 in the young mouse lungs. However, old mouse lungs contain a lower percentage of subset AEC2-1 cells, intact AEC2, and more AEC2-2 and AEC2-3 cells, injured AEC2s relative to the cells from young mouse lungs, suggesting a chronic AEC2 injury and turnover in aged lungs despite no additional overt exogenous injury. This finding is consistent with a pro-inflammatory signature in aged lung (*Angelidis et al., 2019*). During recovery after bleomycin injury, AEC2s in the aged lungs failed to transition effectively back from subsets AEC2-2 and AEC2-3 to subset AEC2-1 cells as occurred in the lungs of young mice. The failure of damaged AEC2s to revert back to intact AEC2s could be associated with impaired progenitor renewal of AEC2s in aged mouse lung. We have reported that AEC2s from aged mouse lungs had decreased renewal capacity relative to AEC2s from young mice due to downregulated sirtuin signaling in aged AEC2s (*Liang et al., 2022*).

In human lungs, we found that AEC2s from healthy lungs have a similar gene signature to the subset AEC2-1 in mice that highly express canonical AEC2 marker genes and are the major AEC2 subset in uninjured lung. Interestingly, IPF AEC2s exhibited a similar gene signature to subsets AEC2-2 and AEC2-3 in injured old mouse lungs. These data suggest that the subset classification of AEC2s we employed in the mouse model has biologic relevance to human disease at least at the gene expression level.

Previous studies showed that CLDN4-expressing cells; KRT8, KRT17, and KRT19 expressing AEC2s acummulated in mouse lungs after injury and in human IPF lungs. These cells were identified as transitional AEC2s and could not effectively differentiate into AEC1s (*Adams et al., 2020*; *Jiang et al., 2020b*; *Kobayashi et al., 2020*; *Strunz et al., 2020*). The subset AEC2-3 we defined in this study should include the transitional AEC2s identified previously based on their elevated claudin family and keratin family gene expression. Our data further suggested that aging enhanced the gene expression of the aberrant AEC2s after lung injury. RNA trajectory analysis of the lineage differentiation potentials suggests that AEC2-3 cells were from subset AEC2-1 via subset AEC2-2. Interestingly, subset AEC2-3 also showed heterogeneity within the cluster, and some cells within this subset were further differentiated into AEC1 cells. However, there is no clear evidence showing dedifferentiation of AEC1 into subset AEC2-3. Careful lineage tracing studies are needed to confirm these transitions. Furthermore, with 3D organoid cultures, we demonstrated that the Cldn4+ subset AEC2-3 cells had impaired progenitor renewal. Our findings provide insights into understanding the role of aging in AEC2 cellular responses to lung injury and repair. It is interesting to speculate that subsets AEC2-2 and AEC2-3 cells may have an enhanced capacity to promote fibrogenesis and future studies are needed to address this hypothesis.

The observation that IPF AECs have a similar gene expression pattern to injured aged mouse AEC2s supports the longstanding concept that repetitive injury of AEC2 cells is a critical component of IPF pathogenesis. What remains a complete conundrum is what is the source of injury in IPF as no exogenous injuriant has ever been identified. The gene expression patterns we identified may yield some clues as how to pursue endogenous sources of injury that could be remediable. The bleomycin mouse model has been widely used as an animal model for human IPF. Our data indicate that bleomycin-injured aged mice might better represent aspects of human disease as it was suggested in a previous study by *Hecker et al., 2014*.

In summary, we have performed comprehensive scRNA-seq analyses of lung alveolar epithelial cells in mouse and man under homeostatic conditions and following lung injury in both old and young mice and in human disease. We identified distinct AEC2 subsets following injury and examined the impact of aging on AEC2 injury and recovery. We uncovered a common theme in gene expression between injured aged mouse AEC2s and human IPF AEC2s. These data suggest that aging is a critical

factor affecting AEC2s response to lung injury and their post injury recovery. Efforts to restore the genetic programs in AEC2s with pharmaceutical reagents might offer new opportunities for therapeutic approaches to aging-associated lung diseases such as IPF.

# Materials and methods

**Key resources table**

| Reagent type (species) or resource | Designation | Source or reference | Identifiers | Additional information |
|---|---|---|---|---|
| Strain, strain background (*Mus musculus*) | C57Bl/6J | Jackson Laboratory | Strain #: 000664 RRID: IMSR_JAX:000664 | |
| Chemical compound, drug | Bleomycin | Hospira, Lake Forest, IL 60045 | NDC 61703-332-18 | 2.5 U/kg in vivo, mice |
| Antibody | Anti-Mouse EpCAM clone G8.8 (rat monoclonal) | BioLegend | Catalog # 118216 RRID: AB_1236471 | Flow 1:200 |
| Antibody | Anti-mouse CD24 clone M1/69 (rat monoclonal) | eBioscience | Catalog # 12-0242-82, RRID: AB_467169 | Flow 1:50 |
| Antibody | Anti-Sca-1 (Ly-6A/E)- clone D7 (rat monoclonal) | eBioscience | Catalog # 17-5981-82, RRID: AB_469487 | Flow 1:200 |
| Antibody | Anti-Mouse CD31 (PECAM-1) clone 390 (rat monoclonal) | eBioscience | Catalog # 13-0311-85, RRID: AB_466421 | Flow 1:40 |
| Antibody | Anti-Mouse CD34 clone RAM34 (rat monoclonal) | eBioscience | Catalog # 13-0341-85, RRID: AB_466425 | Flow 1:16 |
| Antibody | Anti-Mouse CD45 clone 30-F11 (rat monoclonal) | eBioscience | Catalog # 13-0451-85, RRID: AB_466447 | Flow 1:200 |
| Antibody | Anti-human CD31 clone WM59 (mouse monoclonal) | BioLegend | Clone WM59; RRID: AB_314327 | Flow 1:40 |
| Antibody | Anti-human CD45 clone WI30 (mouse monoclonal) | BioLegend | Catalog # 304016, RRID: AB_314404 | Flow 1:200 |
| Antibody | Anti-human EpCAM clone 9C4 (mouse monoclonal) | BioLegend | Catalog # 324212, RRID: AB_756086 | Flow 1:200 |
| Antibody | anti-human Claudin 4 IgG (rabbit polyclonal) | ProteinTech | 16195-1-AP, RRID: AB_2082969 | Flow 1:50; IF 1:200 |
| Antibody | anti-HT2-280 IgM (mouse monoclonal) | Terrace Biotech | TB-27AHT2-280, RRID: AB_2832931 | Flow 1:60; IF 1:200 |
| Sequence-based reagent | mouse *Sftpc*_F | This paper | PCR primers | GCAGGTCCCAGGAGCCAGTTC |
| Sequence-based reagent | mouse *Sftpc*_R | This paper | PCR primers | GGAGCTGGCTTATAGGCCGTCAG |
| Sequence-based reagent | mouse *Pdpn*_F | This paper | PCR primers | GCACCTCTGGTACCAACGCAGA |
| Sequence-based reagent | mouse *Pdpn*_R | This paper | PCR primers | TCTGAGGTTGCTGAGGTGGACAGT |
| Cell line (*Mus musculus*) | MLg2908, lung fibroblast (normal) | ATCC | Catalog CCL-206 | |
| Software, algorithm | Flow Jo | Tree Star | Version 9.9.6 | |

*Continued on next page*

*Continued*

| Reagent type (species) or resource | Designation | Source or reference | Identifiers | Additional information |
|---|---|---|---|---|
| Software, algorithm | RStudio | RStudio PBC | RRID: SCR_000432, version 2022.07.2 build 576 | |
| Software, algorithm | Prism | GraphPad | RRID: SCR_002798, version 8.4.3 | |

## Animals and study approval

All mouse maintenance and procedures were done under the guidance of the Cedars-Sinai Medical Center Institutional Animal Care and Use Committee (IACUC008529) in accordance with institutional and regulatory guidelines. All mice were housed in a pathogen-free facility at Cedars-Sinai. Eight- to 12-week-old (young) and 18- to 24-month-old (aged) wild-type C57Bl/6J mice were obtained from The Jackson Laboratory and housed in the institution facility at least 2 weeks before experiments.

## Information of human subjects, human lung tissue, and study approval

The use of human tissues for research was approved by the Institutional Review Board (IRB) of Cedars-Sinai and was under the guidelines outlined by the IRB (Pro00032727). Informed consent was obtained from each subject. The human samples used in the studies are age matched between IPF and healthy donors. The median age is 60 for healthy donors and 66 for IPF patients. We are aware to get the best age-matched samples within each experiment.

## Bleomycin instillation

Bleomycin instillation (2.5 U/kg) by intratracheal administration was described previously (*Liang et al., 2016*).

## Mouse lung dissociation and flow cytometry

Mouse lung single-cell suspensions and the procedures of staining the cells for flow cytometry and data analysis were described previously (*Barkauskas et al., 2013*; *Chen et al., 2012*; *Liang et al., 2016*). Flow cytometry was performed using a LSRFortessa flow cytometer and FACSAria III sorter (BD, San Jose, CA) and the data were analyzed using Flow Jo 9.9.6 software (Tree Star, Ashland, OR). Primary antibodies EpCAM-PE-Cy7 (clone G8.8, Catalog # 118216, RRID AB_1236471) were from BioLegend. CD24-PE (clone M1/69, Catalog # 12-0242-82, RRID AB_467169), Sca-1 (Ly-6A/E)-APC (clone D7, Catalog # 17-5981-82, RRID AB_469487), CD31 (PECAM-1) (clone 390, Catalog # 13-0311-85, RRID AB_466421), CD34 (clone RAM34, Catalog # 13-0341-85, RRID AB_466425), and CD45 (clone 30-F11, Catalog # 13-0451-85, RRID AB_466447) were all from eBioscience (San Diego, CA). Streptavidin-APC-Cy7 (catalog # 405208) was from BioLegend (San Diego, CA).

## Human lung dissociation and flow cytometry

Human lung single-cell isolation and flow cytometer analysis and sorting were performed as described previously (*Liang et al., 2016*). Anti-human CD31 (clone WM59, Catalog # 303118, RRID AB_2247932), CD45 (clone WI30, Catalog # 304016, RRID AB_314404), and EpCAM (clone 9C4, Catalog # 324212, RRID AB_756086) were from BioLegend.

## Human lung section immunofluorescence staining

Cryosections and immunostaining were following standard protocols. Primary antibodies rabbit anti-Claudin 4 IgG polyclonal antibody (16195-1-AP, RRID AB_2082969, Proteintech) and mouse anti-HT2-280 IgM monoclonal antibody (TB-27AHT2-280, RRID AB_2832931, Terrace Biotech) were used and followed by fluorescence-labeled secondary antibodies.

## scRNA-seq and data analysis

scRNA-seq was performed in Genomics Core at Cedars-Sinai. Flow sorted human and mouse single cells were lysed, and mRNA was reverse transcribed and amplified as previously described (*Liu et al., 2021*). The barcoded libraries were sequenced with NextSeq500 (Illumina, San Diego, CA) to obtain a sequencing depth of ~200K reads per cell.

Raw scRNA-seq data were aligned to human genome GRCh38 and mouse genome mm10 with Cell Ranger, respectively. Downstream quality control, normalization and visualization were performed with Seurat package (*Butler et al., 2018*). For quality control, the output expression matrix from Cell Ranger was done based on number of genes detected in each cell, number of transcripts detected in each cell, and percentage of mitochondrial genes. The expression matrix was then normalized and visualized with UMAP. Bioinformatics analysis of scRNA-seq data was detailed in our recent report (*Liu et al., 2021*). Silhouette analysis of *k*-means clustering was used to assess the separation distance between the resulting clusters. SCRAT was used to determine and envision high-dimensional meta-gene sets exhibited in AEC2 subsets as we described previously (*Liu et al., 2021*; *Xie et al., 2018*). IPA was performed as described previously (*Liu et al., 2021*). The activation score reflects the sum of expression levels of a biological process (or pathway) related genes in each single cell, the results of each cluster cell were shown in Violin plots (*Yao et al., 2021*). The genes in the biological process (or pathway) were downloaded from UniProt. RNA trajectory analysis was performed with Slingshot (*Street et al., 2018*).

## RNA analysis

RNA was extracted from mouse AEC2s using TRIzol Reagent. For real-time PCR analysis, 0.5 µg total RNA was used for reverse transcription with the High Capacity cDNA Reverse Transcription Kit (Applied Biosystems). One microliter cDNA was subjected to real-time PCR by using Power SYBR Green PCR Master Mix (Applied Biosystems) and the ABI 7500 Fast Real-Time PCR system (Applied Biosystems). The specific primers were designed based on cDNA sequences deposited in the GenBank database: mouse *Sftpc* (NM_011359.2), forward 5′-GCAGGTCCCAGGAGCCAGTTC-3′ and reverse 5′-GGAG CTGGCTTATAGGCCGTCAG-3′; mouse *Pdpn* (NM_010329.3), forward 5′-GCACCTCTGGTACCAA CGCAGA-3′ and reverse 5′-TCTGAGGTTGCTGAGGTGGACAGT-3′.

## 3D Matrigel culture of human and mouse AEC2s

Flow sorted mouse (EpCAM$^+$CD31$^-$CD34$^-$CD45$^-$CD24$^-$Sca-1$^-$) AEC2s ($3 \times 10^3$) were cultured in Matrigel/medium (1:1 in volume) mixture in the presence of lung fibroblasts MLg2908 cells ($2 \times 10^5$, Catalog CCL-206, ATCC, Manassas, VA) (*Liang et al., 2016*). MLg2908 has been authenticated using STR profiling and tested free of mycoplasma contamination. Fresh medium with proper treatment was changed every other day. Colonies were visualized with a Zeiss Axiovert40 inverted fluorescent microscope (Carl Zeiss AG, Oberkochen, Germany). Number of colonies with a diameter of ≥50 µm were counted and CFE was determined by the number of colonies in each culture as a percentage of input epithelial cells at 12 days after plating. The treatments were repeated at least three times and similar results were obtained.

## Statistics

The statistical difference between groups in the bioinformatics analysis was calculated using the Wilcoxon signed-rank test. For the scRNA-seq data, the lowest p-value calculated in Seurat was p < $2.2e{-}10^{-16}$. For cell treatment data, the statistical difference between groups was calculated using Prism (version 8.4.3) (GraphPad, San Diego, CA). Data are expressed as the mean ± standard error of the mean. Differences in measured variables between experimental and control group were assessed by using Student's *t*-tests. Results were considered statistically significant at p < 0.05.

## Acknowledgements

The authors thank the members of Noble and Jiang laboratory for support and helpful discussion during the course of the study. This work was supported by National Institutes of Health grants R35-HL150829, R01-HL060539, R01-AI052201 (PWN), and R01-HL122068 (DJ and PWN), R0-1AG078655 (JL and PWN) and P01-HL108793 (PWN and DJ). We thank Dr. L Dobbs of UCSF providing antibodies for the study. The authors thank Genomics Core and Flow Cytometry Core at Cedars for their technical support. Funding information: This work was supported by NIH grants R0-1AG078655, R35-HL150829, R01-HL060539, R01-AI052201, R01-HL122068, and P01-HL108793. Data availability: The single-cell RNA-seq datasets and analysis code files are deposited in GEO and Github, respectively.

# Additional information

## Competing interests

Paul W Noble: Senior editor, *eLife*. The other authors declare that no competing interests exist.

## Funding

| Funder | Grant reference number | Author |
|--------|------------------------|--------|
| National Institute on Aging | R0-1AG078655 | Jiurong Liang |
| National Heart, Lung, and Blood Institute | R35-HL150829 | Paul W Noble |
| National Heart, Lung, and Blood Institute | R01-HL060539 | Paul W Noble |
| National Heart, Lung, and Blood Institute | P01-HL108793 | Dianhua Jiang |

The funders had no role in study design, data collection, and interpretation, or the decision to submit the work for publication.

## Author contributions

Jiurong Liang, Dianhua Jiang, Conceptualization, Resources, Data curation, Formal analysis, Supervision, Funding acquisition, Validation, Investigation, Visualization, Methodology, Writing – original draft, Project administration, Writing – review and editing; Guanling Huang, Resources, Data curation, Software, Formal analysis, Validation, Investigation, Visualization, Methodology, Writing – original draft, Project administration, Writing – review and editing; Xue Liu, Data curation, Software, Formal analysis, Validation, Investigation, Visualization, Methodology; Ningshan Liu, Forough Taghavifar, Formal analysis, Investigation, Methodology; Kristy Dai, Formal analysis, Investigation; Changfu Yao, Data curation, Formal analysis, Investigation, Methodology; Nan Deng, Software, Formal analysis, Investigation, Visualization; Yizhou Wang, Resources, Data curation, Software, Formal analysis, Investigation, Methodology; Peter Chen, Cory Hogaboam, Formal analysis, Writing – review and editing; Barry R Stripp, William C Parks, Resources, Formal analysis, Writing – review and editing; Paul W Noble, Conceptualization, Resources, Formal analysis, Supervision, Funding acquisition, Investigation, Visualization, Methodology, Writing – original draft, Project administration, Writing – review and editing

## Author ORCIDs

Dianhua Jiang http://orcid.org/0000-0002-4508-3829

## Ethics

Information of human subjects, human lung tissue, and study approval. The use of human tissues for research was approved by the Institutional Review Board (IRB) of Cedars-Sinai and was under the guidelines outlined by the IRB (Pro00032727). Informed consent was obtained from each subject. The human samples used in the studies are age matched between IPF and healthy donors. The median age is 60 for healthy donors and 66 for IPF patients. We are aware to get the best age-matched samples within each experiment.

Animals and study approval. All mouse maintenance and procedures were done under the guidance of the Cedars-Sinai Medical Center Institutional Animal Care and Use Committee (IACUC008529) in accordance with institutional and regulatory guidelines. All mice were housed in a pathogen-free facility at Cedars-Sinai. Eight- to 12-week-old (young) and 18- to 24-month-old (aged) wild-type C57Bl/6J mice were obtained from The Jackson Laboratory and housed in the institution facility at least 2 weeks before experiments.

## Decision letter and Author response

Decision letter https://doi.org/10.7554/eLife.85415.sa1
Author response https://doi.org/10.7554/eLife.85415.sa2

## Additional files

### Supplementary files

• Supplementary file 1. Significantly enriched canonical pathways in AEC2 subsets using Ingenuity Pathway Analysis (IPA). (**a**) Ingenuity pathway analysis showing activation *Z*-scores for selected pathways in AEC2-2 compared to AEC2-1. (**b**) Ingenuity pathway analysis showing activation *Z*-scores for selected pathways in AEC2-3 compared to AEC2-1.

• MDAR checklist

### Data availability

The raw datasets of single-cell RNA-seq of mouse and human epithelial cells are under GSE157995 and GSE157996, respectively.

The following datasets were generated:

| Author(s) | Year | Dataset title | Dataset URL | Database and Identifier |
|---|---|---|---|---|
| Huang G, Liang J, Noble PW, Jiang D | 2022 | Single-cell RNA-sequencing of bleomycin-injured young and old mice | https://www.ncbi.nlm.nih.gov/geo/query/acc.cgi?acc=GSE157995 | NCBI Gene Expression Omnibus, GSE157995 |
| Huang G, Liang J, Noble PW, Jiang D | 2022 | Single-cell RNA-sequencing of human lung Lin-EpCAM+ cells | https://www.ncbi.nlm.nih.gov/geo/query/acc.cgi?acc=GSE157996 | NCBI Gene Expression Omnibus, GSE157996 |

The following previously published dataset was used:

| Author(s) | Year | Dataset title | Dataset URL | Database and Identifier |
|---|---|---|---|---|
| Liang J, Huang G, Liu X, Taghavifar F | 2022 | Single-cell RNA-sequencing of mouse and human lung Lin-EpCAM+ cells | https://www.ncbi.nlm.nih.gov/geo/query/acc.cgi?acc=GSE157997 | NCBI Gene Expression Omnibus, GSE157997 |

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
