## [Editor Report]

These findings demonstrate the relationship between lung injury, aging, and lung fibrosis with strong evidence that aging modulates the response to lung injury and wound healing. This work furthers our understanding of fibrosis in humans, where idiopathic pulmonary fibrosis occurs most commonly at an advanced age.

---

## [Decision Letter]

**Decision letter after peer review:**

Thank you for submitting your article "Reciprocal interactions between alveolar progenitor dysfunction and aging promote lung fibrosis" for consideration by *eLife*. Your article has been reviewed by 2 peer reviewers, and the evaluation has been overseen by a Reviewing Editor and Mone Zaidi as the Senior Editor. The following individuals involved in review of your submission have agreed to reveal their identity: Victor J Thannickal (Reviewer #1).

Essential revisions:

1) With regard to the intermediate "transitional" states, the authors might consider performing RNA trajectory analysis to better define the dynamic changes and relationship between transcriptional states. For example, do the presumed transitional cells that express CLDN4 and keratin isoforms derive from healthy (young) AECs, or is it possible that AEC1 de-differentiation gives rise to these aberrant intermediate states? How does aging shift the trajectories of AECs in response to lung injury?

2) The identification of 3 distinct sub-types of AEC2s, which shift w/age and in bleo-injury is of interest and may yield future targetable genes for a range of age- or injury-related processes. The lack of proof-of-concept or mechanisms in terms of demonstrating that the 3 sub-types have differential roles in promoting (or stopping) lung fibrosis confers a descriptive status to this manuscript. However, if the goal of this manuscript is to present initial single-cell findings to the public using mouse and human samples, then what is presented is sufficient.

*Reviewer #1 (Recommendations for the authors):*

The authors employ single cell RNA sequencing of isolated alveolar type 2 cells (AEC2) to identify three clustered subsets that are influenced by age and lung injury. A synergistic effect of aging and lung injury contributed to impaired AEC2 recovery in aged mice after injury. Additionally, AEC2s from human idiopathic pulmonary fibrosis (IPF) revealed a similar transcriptomic signature to AEC2 subsets from bleomycin injured lungs from old mice. The studies are well done and provide new insights into how aging modulates the lung injury response in AECs.

With regard to the intermediate "transitional" states, the authors might consider performing RNA trajectory analysis to better define the dynamic changes and relationship between transcriptional states. For example, do the presumed transitional cells that express CLDN4 and keratin isoforms derive from healthy (young) AECs, or is it possible that AEC1 de-differentiation gives rise to these aberrant intermediate states? How does aging shift the trajectories of AECs in response to lung injury?

*Reviewer #2 (Recommendations for the authors):*

This manuscript examines genomic programs of alveolar type 2 cells (AEC2s) in uninjured and bleomycin-injured young and old mice. Three subsets of AEC2s were identified based on their single-cell gene signatures, which were correlated with processes such as progenitor cell renewal, inflammation, stress and aging. A human correlation was made from human lungs. The conclusion was that new insights into how aging and lung injury interact and synergize to potentially promote fibrosis.

The use of unbiased approaches (unclustered, single-cell RNA technology) is appropriate and innovative, as well as informative. The parallel studies of mouse and human lungs provides impactful, substantive credibility to the data. The data support the original hypothesis. Therefore, this is considered an important and impactful study.

The background literature is well covered, concise and the methodologies rigorous. The identification of 3 distinct sub-types of AEC2s, which shift w/age and in bleo-injury is of interest and may yield future targetable genes for a range of age- or injury-related processes. The lack of proof-of-concept or mechanisms in terms of demonstrating that the 3 sub-types have differential roles in promoting (or stopping) lung fibrosis confers a descriptive status to this manuscript. However, if the goal of this manuscript is to present initial single-cell findings to the public using mouse and human samples, then what is presented is sufficient.

---

## [Author Response]

Essential revisions:Reviewer #1 (Recommendations for the authors):The authors employ single cell RNA sequencing of isolated alveolar type 2 cells (AEC2) to identify three clustered subsets that are influenced by age and lung injury. A synergistic effect of aging and lung injury contributed to impaired AEC2 recovery in aged mice after injury. Additionally, AEC2s from human idiopathic pulmonary fibrosis (IPF) revealed a similar transcriptomic signature to AEC2 subsets from bleomycin injured lungs from old mice. The studies are well done and provide new insights into how aging modulates the lung injury response in AECs.With regard to the intermediate "transitional" states, the authors might consider performing RNA trajectory analysis to better define the dynamic changes and relationship between transcriptional states. For example, do the presumed transitional cells that express CLDN4 and keratin isoforms derive from healthy (young) AECs, or is it possible that AEC1 de-differentiation gives rise to these aberrant intermediate states? How does aging shift the trajectories of AECs in response to lung injury?

We thank the reviewer for the suggestion. We performed RNA trajectory analysis using Slingshot. The lineage differentiation potentials suggests that AEC2-3 cells were from subset AEC2-1 via subset AEC2-2. Interestingly, subset AEC2-3 also showed heterogeneity within the cluster, and some cells within this subset were further differentiated into AEC1 cells. The significance of AEC2-3 heterogeneity is unknown. There was slightly difference in the trajectory of AEC2-3 between young and old, possibly due to the difference in heterogeneity of AEC2-3. However, there is no clear evidence showing dedifferentiation of AEC1 into subset AEC2-3. Of course, careful lineage tracing studies are needed to confirm these transitions.

These data are added as Figure 2E.

Reviewer #2 (Recommendations for the authors):This manuscript examines genomic programs of alveolar type 2 cells (AEC2s) in uninjured and bleomycin-injured young and old mice. Three subsets of AEC2s were identified based on their single-cell gene signatures, which were correlated with processes such as progenitor cell renewal, inflammation, stress and aging. A human correlation was made from human lungs. The conclusion was that new insights into how aging and lung injury interact and synergize to potentially promote fibrosis.The use of unbiased approaches (unclustered, single-cell RNA technology) is appropriate and innovative, as well as informative. The parallel studies of mouse and human lungs provides impactful, substantive credibility to the data. The data support the original hypothesis. Therefore, this is considered an important and impactful study.The background literature is well covered, concise and the methodologies rigorous. The identification of 3 distinct sub-types of AEC2s, which shift w/age and in bleo-injury is of interest and may yield future targetable genes for a range of age- or injury-related processes. The lack of proof-of-concept or mechanisms in terms of demonstrating that the 3 sub-types have differential roles in promoting (or stopping) lung fibrosis confers a descriptive status to this manuscript. However, if the goal of this manuscript is to present initial single-cell findings to the public using mouse and human samples, then what is presented is sufficient.

We thank the reviewer for the comments. We agree that further studies are needed to comprehensively demonstrate the roles and mechanisms of AEC2 subsets in aging and lung fibrosis. The study provided important resources to the research community.